# Feature selection for high dimensional microarray gene expression data via weighted signal to noise ratio

Muhammad Hamraz[1☯], Amjad Ali[1☯], Wali Khan Mashwani[2☯], Saeed Aldahmani[3☯], Zardad Khan[3☯]*

1 Department of Statistics, Abdul Wali Khan University Mardan, Mardan, Pakistan, 2 Institute of Numerical Sciences, Kohat University of Science and Technology, Kohat, Pakistan, 3 Department of Analytics in the Digital Era, United Arab Emirates University, Al Ain, UAE

☯ These authors contributed equally to this work.
* zaar@uaeu.ac.ae

## Abstract

Feature selection in high dimensional gene expression datasets not only reduces the dimension of the data, but also the execution time and computational cost of the underlying classifier. The current study introduces a novel feature selection method called weighted signal to noise ratio ($W_{SNR}$) by exploiting the weights of features based on support vectors and signal to noise ratio, with an objective to identify the most informative genes in high dimensional classification problems. The combination of two state-of-the-art procedures enables the extration of the most informative genes. The corresponding weights of these procedures are then multiplied and arranged in decreasing order. Larger weight of a feature indicates its discriminatory power in classifying the tissue samples to their true classes. The current method is validated on eight gene expression datasets. Moreover, results of the proposed method ($W_{SNR}$) are also compared with four well known feature selection methods. We found that the ($W_{SNR}$) outperform the other competing methods on 6 out of 8 datasets. Box-plots and Bar-plots of the results of the proposed method and all the other methods are also constructed. The proposed method is further assessed on simulated data. Simulation analysis reveal that ($W_{SNR}$) outperforms all the other methods included in the study.

## 1 Introduction

Feature/Gene selection in micro-array gene expression datasets has gained great attention during the recent decades [1–7]. Since high dimensional datasets usually contain noisy, redundant and non-informative features that enhance computational complexity as well as execution time of the underlying model. Feature selection is therefore, necessary to select the informative features and remove the unnecessary ones. This will not only reduce execution or training time but will also increase the accuracy of the model. Based on this model one can categorize the samples in the data into their classes [8]. Feature selection is mainly carried out by using three different methods such as wrapper, filter and embedded. The feature selection methods

**Competing interests:** The authors have declared that no competing interests exist.

used in paper falls under the category of filter methods, except sigF [9] which is a wrapper method. Features or variables selection is used in variety of task such as classification, regression and clustering [10]. Also, different types of biological data sets can be analyzed by using feature selection, for instance whole-genome sequencing data set [11], protein mass spectra data set [12], whole-genome expression data set [13–15], and so on. Micro-array and other high throughput technologies are capable of measuring thousands of genes simultaneously, leading to its rampant usage in clinical settings. Recent years have witnessed a lot of feature selection methods for micro-array data analysis. Authors in [16] introduced a method known as 'double feature selection method'. In their method they have used both the global and intrinsic geometric information, for the selection of informative features in data. Similarly, study in [17] introduced a method that handles semi-supervised feature selection tasks. This method combines neighborhood discriminant index (NDI) and forward iterative Laplacian score (FILS) methods for the selection of discriminative features in high-dimensional data sets. A more efficient implementation of linear support vector machines to improve the recursive feature elimination strategy and then combine them together to select informative genes was proposed in [18]. A study in [2] proposed a new technique that applies an ensemble of feature selection procedures to select those genes that are highly correlated to Lung Adenocarcinoma (LUAD). Utilizing LUAD RNA-seq data from the Cancer Genome Atlas (TCGA), mutual information (MI) was employed followed by recursive feature elimination (RFE) feature selection procedures along with SVM classifier. A new Bi-dimensional Principal Feature Selection (BPFS) procedure for efficiently extracting critical genes was proposed for high dimensional gene expression datasets [19]. This procedure utilizes the principal component analysis (PCA) technique on sample and gene domains successively, in order to identify the informative genes and reduce redundancies while losing less information. The selection of informative features and their importance in classification/regression can be found in [20–27]. The main focus of these methods is to enhance the classification accuracy of the underlying classifier with the help of selected genes, while ignoring their biological relevance, which leads to inaccurate downstream data analysis [28–33]. Therefore, it is necessary to device such a feature selection method that not only increase the classification accuracy, but also to be capable of identifying the biological significance of the selected genes, in tumor versus normal tumor contrast [34, 35].

This paper proposes a new feature selection procedure by combining the information obtained from well known feature selection method called signal to noise ratio (SNR) [40] and the feature weights given by support vector machine (SVM) [36]. For assessing the performance of the current study eight gene expression datasets, i.e., Leukemia, Colon, Srbct, DLBCL, Lungcancer, Breastcancer, TumorC and Prostate have been used. Furthermore, the results of the proposed method are compared with four other well known feature selection methods such as significant features "sigF" [9], minimum redundancy maximum relevance (mRmR) [37], wilcoxon rank sum test "Wilc" [38] and an ensemble method called SVM-mRMRe [39]. After comparing the results of the proposed method ($W_{SNR}$) with the aforemesioned methods, it has been observed that the proposed $W_{SNR}$ stands apart in terms of classification error. Box-plots and bar-plots of the results are also constructed, which also indicate that the proposed method has better performance as compared to the aforementioned feature selection methods. The rest of the paper is organized as follows.

Section 2 gives a detailed description of the datasets used in the paper, support vector machine (SVM) classifier, feature selection procedures "Significant Features" (sigF) [9], Signal to Noise Ratio (SNR) [40], and the proposed method ($W_{SNR}$) with its mathematical background and algorithm. Section 3 presents the experimental set up of the proposed method.

**Table 1. Brief description of the datasets along with the corresponding number of features, observations, class-wise distributions and sources.**

| Data set | Genes | Samples | Class Distribution | Sources |
|---|---|---|---|---|
| Leukemia | 7129 | 72 | (49, 23) | [41] |
| Colon | 2000 | 62 | (40, 22) | [42] |
| Breast | 4948 | 78 | (34,44) | [43] |
| Srbct | 2808 | 54 | (29, 25) | [44] |
| Lung | 12600 | 148 | (134, 14) | [45] |
| DLBCL | 7070 | 77 | (58, 19) | [46, 47] |
| TumorC | 7129 | 60 | (39, 21) | https://www.openml.org |
| Prostate | 10936 | 413 | (344, 69) | https://www.openml.org |

Section 4 gives discussion on the results of the proposed method $W_{SNR}$. The paper is concluded in Section 5.

## 2 Methods

### 2.1 Data sets

For the assessment of the proposed method, $W_{SNR}$, eight benchmark problems are used. Their sources along with number of features, number of observations and class wise distribution of samples are given in Table 1.

### 2.2 Support vector machine

Support vector machine (SVM) is a supervised learning technique, which has been widely used for regression and classification problems in literature. It has also been used for feature selection in several studies [32, 33, 48]. This classifier utilizes several kernel functions to perform the classification effectively in linear and non-linear feature spaces. The SVM searches a linear or non-linear optimal hyperplane ($H$), which can then divide the two groups of observations meaningfully [49]. This hyperplane ($H$) is supposed to be at maximum distance from both the classes or groups in high-dimensional spaces, so as to separate the two groups as much as possible. The hyperplane is represented in the form of a vector given in Eq 1 which acts as a reference frame to identify the position of each sample or observation in high-dimensional spaces. It is summed in order to produce a score known as discriminate score, which is then used to categorize the observations into one of the two classes.

$$y = w^T(\psi(z)) + b \tag{1}$$

where y is a response vector, i.e., y $\in$ (0, 1), where each sample in the data is classified into class 0 or 1. $z = (z_1, \cdots, z_d)$ is a d-dimensional input vector and vector $w = (w_1, \cdots, w_d)$ contains the coefficients of the hyperplane. The term $b$ indicates the intercept of the hyperplane.

**2.2.1 Mathematical description behind SVM weights $w$.** As the SVM algorithm uses a hyperplane ($H$) to classify the data points in their respective classes, i.e.,

$$H : w^T(\psi(z)) + b = 0$$

The distance between a given point $\psi(z_0)$ and the hyperplane $H$ is give by

$$d_H(\psi(z)) = \frac{\|w^T(\psi(z)) + b\|}{\|w\|_2}, \tag{2}$$

where $\|w\|_2$ is the Euclidean norm given as

$$\|w\|_2 = \sqrt{w_1^2 + w_2^2 + w_3^2 + \ldots + w_d^2}. \tag{3}$$

The weight vector is the argument that maximize the distance given in Eq 2, that is:

$$w = \arg - \max_w [min\{d_H(\psi(z))\}]. \tag{4}$$

## 2.3 Significant feature selection (sigF)

A method known as Signature feature selection (sigF) can be found in [9]. In this method, significant features are identified with the help of support vector machine and t-test. First, the weight of each feature is computed via support vector machine (SVM). In the second stage, t-test is computed for each feature in the data in the following manner:

$$t_j = \frac{\bar{z}_j^0 - \bar{z}_j^1}{\sqrt{((s_j^0)^2/n^0) + ((s_j^1)^2/n^1)}}, \tag{5}$$

where $\bar{z}_j^0, \bar{z}_j^1, s_j^0, s_j^1, n^0, n^1$ represent the means, standard deviations and number of samples in Class 0 and 1, respectively. In this way the $t$-statistic is computed for each feature in the data. Alternatively, $p$-values for all the features in the data are computed based on $t$-test. A smaller p-value of a feature represents its discriminative ability. The weights computed via SVM classifier are then multiplied with these $p$-values to achieve new weights of all the features by using the following equation.

$$\xi_j = w_j * P[t_j(v) < -\|u\|ort_j(v) > \|u\|], \tag{6}$$

where $v$ is the level of significance for the corresponding reference distribution and $u$ is the observed value of test statistic based on the level of flexibility $v$. The feature is considered informative if it possesses a smaller value of $\xi$.

## 2.4 The proposed method, $W_{SNR}$

The proposed method selects the informative genes or features in high-dimensional gene expression data sets in a similar fashion as that of sigF given in [9]. The only difference is that the method in [9] computes $t$-statistic for each feature, which is then multiplied with the weights computed via support vector machine classifier. The proposed method on the other hand computes signal to noise ratio [40] for each feature in the following manner.

$$SNR_j = \left\|\frac{\bar{z}_j^0 - \bar{z}_j^1}{s_j^0 + s_j^1}\right\|, \tag{7}$$

where $\bar{z}_j^0, \bar{z}_j^1, s_j^0, s_j^1$ represent the mean and standard deviations of class 0 and 1, respectively. Features that carry larger value of SNR, are supposed to have greater discriminative ability. Similarly weights of all the features in the data are also computed via SVM, i.e., $w_j$. Since both the weighting schemes assign larger weights to the informative genes therefore, their multiplication will also assign larger weights to the features that are informative. The resultant weights of the proposed method ($W_{SNR}$) are computed by using the following equation

$$(W_{SNR})_j = w_j * SNR_j, \tag{8}$$

where $(W_{SNR})_j$ represents the weight of $j^{th}$ feature in the data. The proposed method ($W_{SNR}$) considers the following steps in identifying the informative genes.

- Compute weights of all the features using support vectors and denote it by $w_j$.

- Compute signal to noise ratio for all the features in the training data and denote it by $SNR_j$.

- Multiply the corresponding weights in step 1 and 2 and arrange them in descending order.

- Select the top ranked ($K$) genes in step 3 for the model construction.

The authors in [9] have used t-test rather than signal to noise ratio for the selection of discriminative genes. The t-test requires the underlying distribution of variables to be approximately normal, which is a difficult task in a situation where data contains tens of thousand of genes or variables. On the other hand signal to noise ratio does not require such assumption. The following pseudo code given in Algorithm 1 explains how the proposed method, $W_{SNR}$, identifies the informative genes, in high-dimensional gene expression data sets, followed by its flowchart in Fig 1.

**Algorithm 1** Pseudo code of the proposed method, $W_{SNR}$.

```
1: M = (X, Y)ₙ×(d+1) ← Micro-array data with dimension n × (d + 1);
2: n ← Number of tissue samples in the data;
3: d ← Number of genes in the data;
```

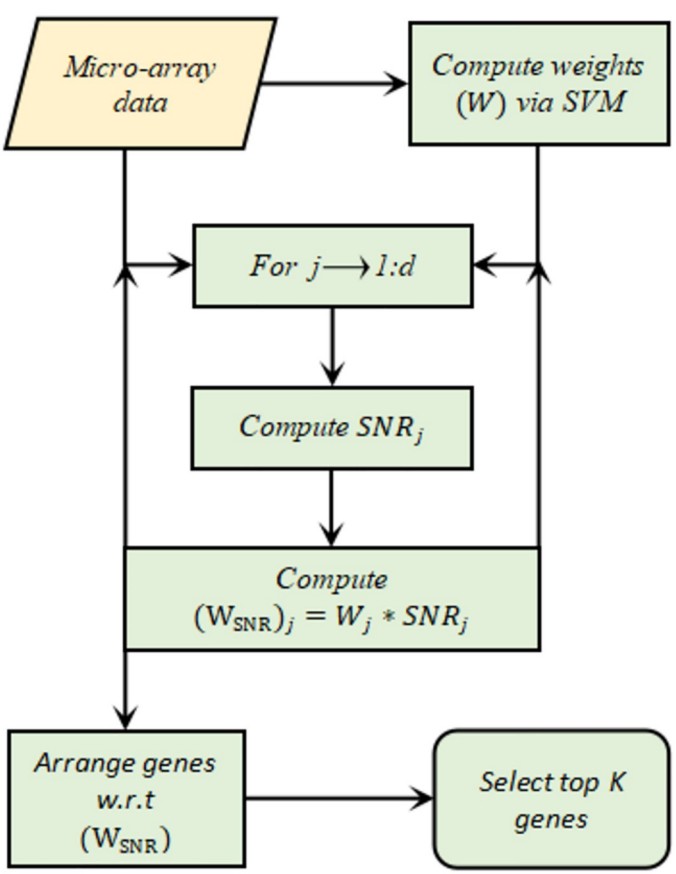

**Fig 1. Flowchart of the proposed method.**

```
4: X_{n×d} ← Total input feature space with n samples and d genes;
5: Y ← Target variable having n values.
6: K ← Number of genes to be selected.
7: w ← Weights vector of genes obtained via support vector;
8: w_j ← Weight of j^{th} gene obtained via support vector;
9: for j ← 1: d do
10:   SNR_j ← Compute the using signal to noise ratio;
11:   Perform (W_{SNR})_j = w_j * SNR_j;
12: end for
13: Arrange the weights (W_{SNR})_j in decreasing order;
14: Select the top K genes for model construction.
```

## 3 Experiments

This section provides the experimental setup of the current paper. Eight high-dimensional gene expression benchmark problems are analyzed, where each benchmark problem is split into (70%) training and (30%) testing parts. This splitting criteria is repeated 500 times for all feature selection procedures and the classifiers used for assessing their performance. Random forest (RF) and k-Nearest Neighbours (k-NN) classifiers have been used to evaluate the performance of different subsets of informative genes selected by various feature selection techniques.

The feature selection method, minimum redundancy and maximum relevance (mRmR), is implemented in R package mRMRe [50]. Wilcoxon rank sum test (Wilc) and significant feature selection (sigF) are implemented by using the R packages WilcoxCV [51] and sigFeature [9], respectively. Moreover, the R library randomForest [52] is utilized for fitting the random forest algorithm with default parameters, i.e., ntree = 500, mtry = $\sqrt{p}$ and nodesize = 1. Similarly, the R library caret [53] is used for the implementation of k-Nearest neighbours classifier, with parameter $k = 5$.

The training parts of each benchmark problem are considered for the selection of different subsets of descriminative genes, i.e., $K = 5$, 10 and 15 by different gene selection procedures to train the classifiers. Classification error rate is used as a performance metric to investigate the classifiers' performance on the basis of selected set of informative genes.

## 4 Results and discussion

Table 2 provides the classification error rates produced by the proposed method, $W_{SNR}$, and all the other competitors included in the study, for different subsets of informative genes. From Table 2, it is evident that for the data set "Leukemia" the proposed method has outperformed all the other methods on both the classifiers. In the case of "Colon" data set, the proposed method has outperformed the others on random forest classifier for all subsets of descriminative genes, while on $k$-nearest neighbour classifier the method (sigF) has produced minimum error for a subset of 5 informative genes. The proposed method, however, has produced minimum error rates for the subsets of genes 10 and 15. Similarly, in the case of "Lungcancer" data set, the method (Wilc) has yielded minimum error rates on random forest classifier while the proposed method has outperformed all the other competitors on $k$-NN classifier. In the case of "Srbct" data set, the proposed $W_{SNR}$ method has outperformed all the other methods except for the number of 5 informative genes, where the method "sigF" has yielded minimum error rate on $k$-NN classifier. The proposed method has outperformed all the other methods on random forest classifier in the case of the dataset "DLBCL" and has shown poor performance on $k$NN classifier. Similarly, the $W_{SNR}$ method has won over all the other procedures in majority of the cases for the data set "Breast" but has shown poor performance in case of "TumorC"

**Table 2. Classification error rates produced by different methods on various subsets of genes.**

| Dataset | Genes | RF | | | | | KNN | | | | |
|---|---|---|---|---|---|---|---|---|---|---|---|
| | | $W_{SNR}$ | sigF | Wilc | mRmR | SVM-mRMRe | $W_{SNR}$ | sigF | Wilc | mRmR | SVM-mRMRe |
| Leukemia | 5 | **0.002** | 0.171 | 0.035 | 0.172 | 0.081 | **0.019** | 0.224 | 0.116 | 0.254 | 0.026 |
| | 10 | **0.004** | 0.173 | 0.024 | 0.090 | 0.112 | **0.025** | 0.237 | 0.129 | 0.255 | 0.028 |
| | 15 | **0.005** | 0.156 | 0.013 | 0.114 | 0.092 | **0.027** | 0.219 | 0.112 | 0.317 | 0.033 |
| Colon | 5 | **0.219** | 0.242 | 0.347 | 0.366 | 0.228 | 0.262 | 0.207 | 0.529 | 0.367 | **0.202** |
| | 10 | **0.182** | 0.254 | 0.306 | 0.326 | 0.194 | **0.224** | 0.237 | 0.495 | 0.344 | 0.228 |
| | 15 | **0.163** | 0.244 | 0.296 | 0.319 | 0.167 | **0.214** | 0.217 | 0.526 | 0.338 | 0.217 |
| Lungcancer | 5 | 0.027 | 0.103 | **0.022** | 0.100 | 0.026 | **0.026** | 0.099 | 0.133 | 0.098 | 0.029 |
| | 10 | 0.040 | 0.095 | **0.000** | 0.097 | 0.003 | **0.040** | 0.101 | 0.111 | 0.108 | 0.041 |
| | 15 | 0.019 | 0.092 | **0.000** | 0.090 | 0.002 | **0.022** | 0.096 | 0.067 | 0.092 | 0.026 |
| Srbct | 5 | **0.047** | 0.059 | 0.137 | 0.339 | 0.048 | 0.064 | **0.050** | 0.072 | 0.288 | 0.053 |
| | 10 | **0.026** | 0.036 | 0.084 | 0.094 | 0.028 | **0.027** | 0.044 | 0.074 | 0.211 | 0.046 |
| | 15 | **0.016** | 0.031 | 0.086 | 0.097 | 0.020 | **0.010** | 0.034 | 0.068 | 0.266 | 0.049 |
| DLBCL | 5 | **0.151** | 0.257 | 0.239 | 0.271 | 0.157 | 0.169 | 0.291 | 0.132 | 0.276 | **0.126** |
| | 10 | **0.138** | 0.252 | 0.216 | 0.257 | 0.141 | 0.156 | 0.342 | 0.144 | 0.283 | **0.131** |
| | 15 | **0.141** | 0.191 | 0.211 | 0.266 | 0.178 | 0.156 | 0.304 | 0.137 | 0.300 | **0.123** |
| Breast | 5 | **0.360** | 0.490 | 0.370 | 0.416 | 0.413 | **0.386** | 0.448 | 0.399 | 0.482 | 0.404 |
| | 10 | **0.364** | 0.514 | 0.371 | 0.458 | 0.433 | **0.372** | 0.501 | 0.394 | 0.413 | 0.384 |
| | 15 | 0.377 | 0.519 | **0.351** | 0.470 | 0.507 | **0.367** | 0.514 | 0.394 | 0.451 | 0.395 |
| TumorC | 5 | 0.438 | 0.482 | 0.424 | **0.373** | 0.381 | 0.438 | 0.407 | **0.390** | 0.481 | 0.392 |
| | 10 | 0.412 | 0.482 | 0.407 | **0.371** | 0.384 | 0.410 | 0.471 | **0.389** | 0.449 | 0.394 |
| | 15 | 0.404 | 0.482 | 0.397 | **0.345** | 0.366 | 0.399 | 0.415 | **0.384** | 0.444 | 0.397 |
| Prostate | 5 | **0.191** | 0.197 | 0.193 | 0.192 | 0.193 | **0.226** | 0.228 | 0.228 | 0.229 | 0.241 |
| | 10 | **0.183** | 0.205 | 0.185 | 0.184 | 0.198 | 0.204 | 0.191 | **0.184** | 0.207 | 0.240 |
| | 15 | **0.163** | 0.186 | 0.166 | 0.179 | 0.189 | **0.175** | 0.200 | 0.177 | 0.190 | 0.214 |

data set. Similarly, the proposed method has won over all the other methods in case of Prostate data set. Overall, the method, $W_{SNR}$, has produced minimum error rates in six out of eight data sets and comparable results on one data set. To summarize simulation results, a win-loss summary is given in Table 3.

The performance of the proposed method is also illustrated with the help of bar-plots of the results for pictorial illustration as given in Figs 2–9. It is clear from the plots that in case of the data set "Leukemia" the heights of bars corresponding to the proposed method, $W_{SNR}$, are

**Table 3. Win-loss table of the methods used.** Total number of wins of the methods on the data sets is given in the last row of the table.

| Datasets | RF | | | | | kNN | | | | |
|---|---|---|---|---|---|---|---|---|---|---|
| | $W_{SNR}$ | sigF | Wilc | mRmR | SVM-mRMRe | $W_{SNR}$ | sigF | Wilc | mRmR | SVM-mRMRe |
| Leukemia | win | loss | loss | loss | loss | win | loss | loss | loss | loss |
| Colon | win | loss | loss | loss | loss | win | loss | loss | loss | loss |
| Lungcancer | loss | loss | win | loss | loss | win | loss | loss | loss | loss |
| Srbct | win | loss | loss | loss | loss | win | loss | loss | loss | loss |
| DLBCL | win | loss | loss | loss | loss | loss | loss | loss | loss | win |
| Breast | win | loss | loss | loss | loss | win | loss | loss | loss | loss |
| TumorC | loss | loss | loss | win | loss | loss | loss | win | loss | loss |
| Prostate | win | loss | loss | loss | loss | win | loss | loss | loss | loss |
| Total win | 6 | 0 | 1 | 1 | 0 | 6 | 0 | 1 | 0 | 1 |

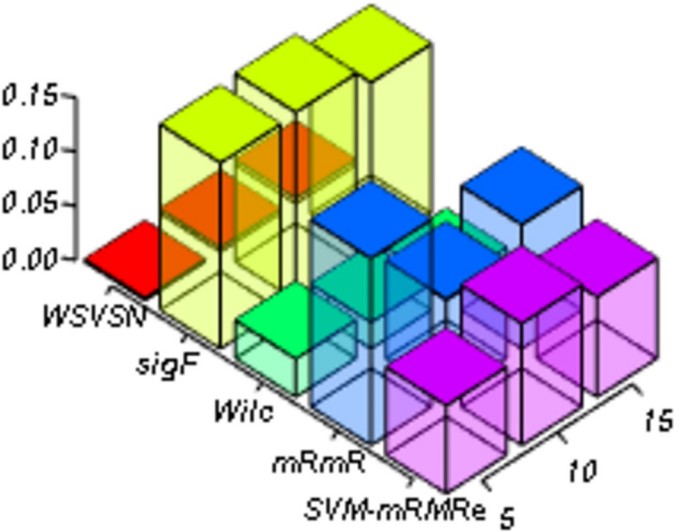

**Fig 2. Bar-plots of error rates of the proposed and the other classical methods on various subsets for Leukemia dataset.**

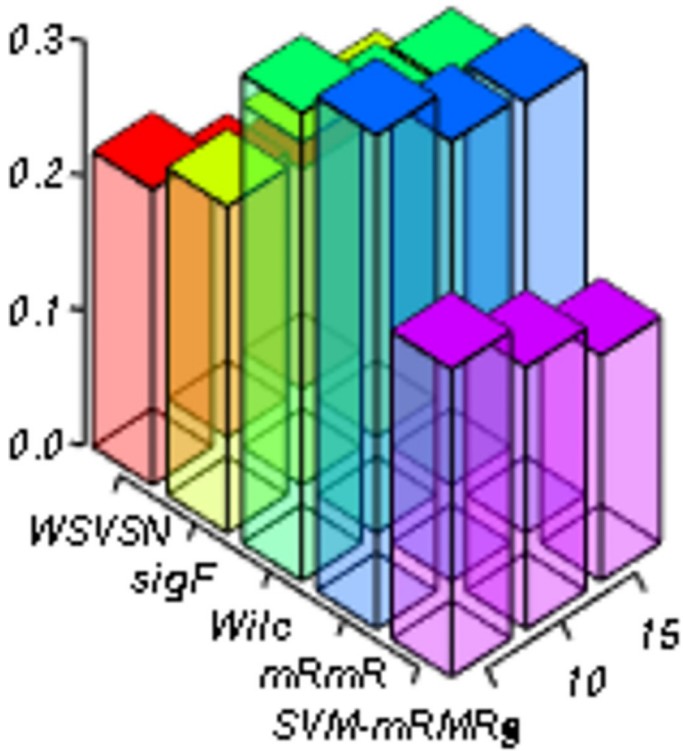

**Fig 3. Bar-plots of error rates of the proposed and the other classical methods on various subsets of genes for Colon dataset.**

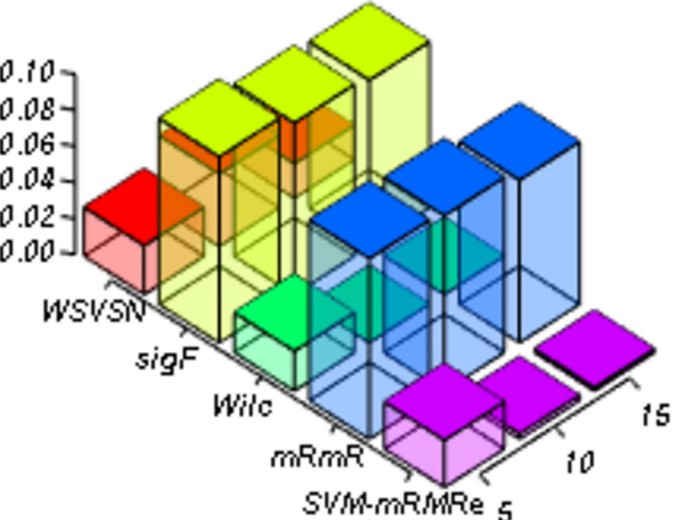

**Fig 4. Bar-plots of error rates of the proposed and the other classical methods on various subsets of genes for Lungcancer dataset.**

smaller than the bars corresponding to all the other procedures included in the study. In case of data set "Lungcancer" the method "Wilc" is producing minimum error rates than the rest of the gene selection procedures. For the data sets "Srbct" and "DLBCL", the method, $W_{SNR}$, method has produced minimum classification error rates. For the remaining data sets, our method has maintained a majority wining position except for the data set "TumorC". Fig 2 has

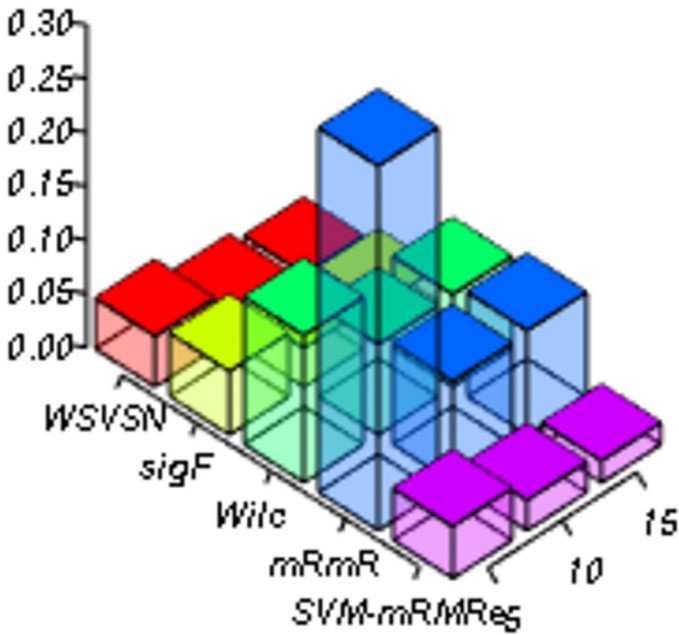

**Fig 5. Bar-plots of error rates of the proposed and the other classical methods on various subsets of genes for Srbct dataset.**

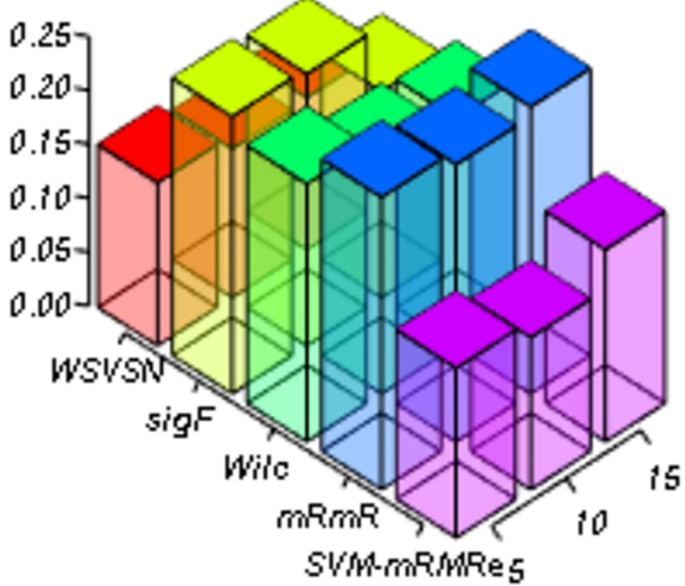

**Fig 6. Bar-plots of error rates of the proposed and the other classical methods on various subsets of genes for DLBCL dataset.**

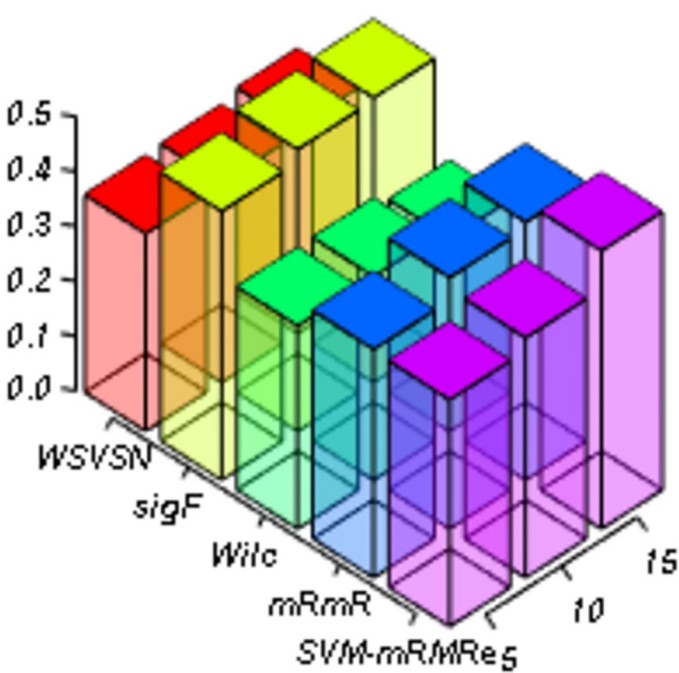

**Fig 7. Bar-plots of error rates of the proposed and the other classical methods on various subsets of genes for Breast dataset.**

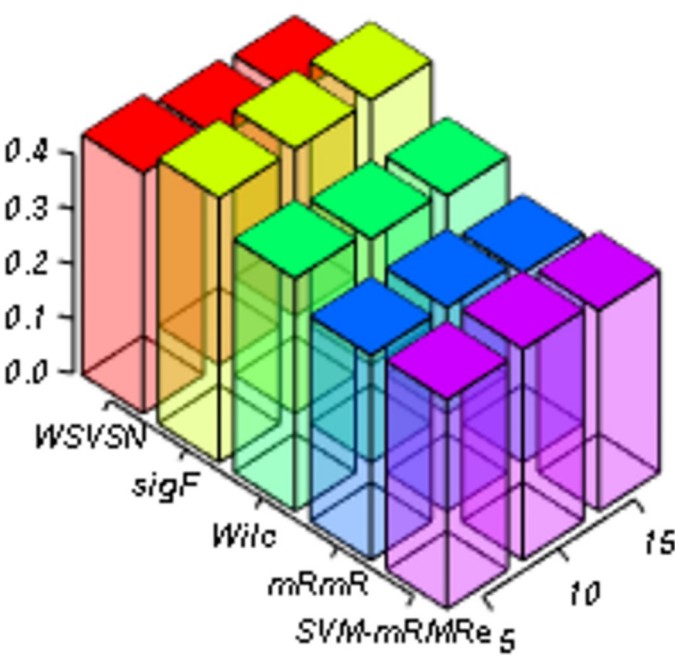

**Fig 8. Bar-plots of error rates of the proposed and the other classical methods on various subsets of genes for TumorC dataset.**

been constructed for a quick insight into the results of various feature selection methods included in the study.

Similarly, box-plots of the results produced by the method, $W_{SNR}$, and all the other competitors for 10 number of informative genes on random forest classifier are also constructed as

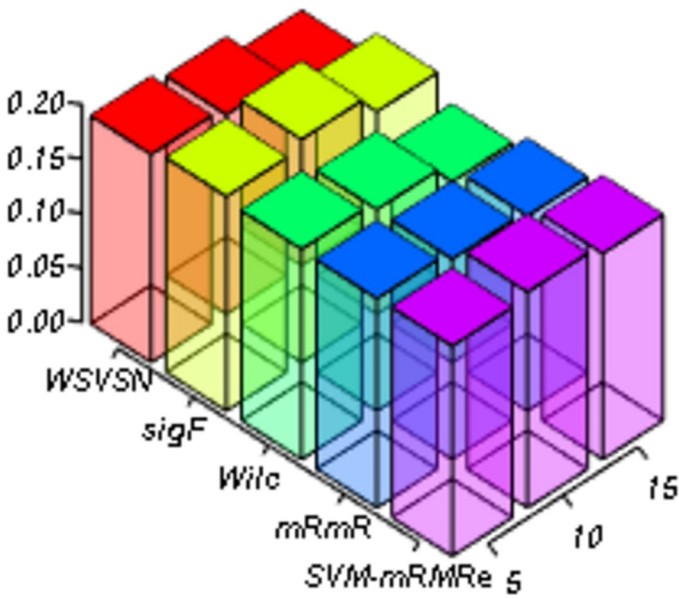

**Fig 9. Bar-plots of error rates of the proposed and the other classical methods on various subsets of genes for Prostate dataset.**

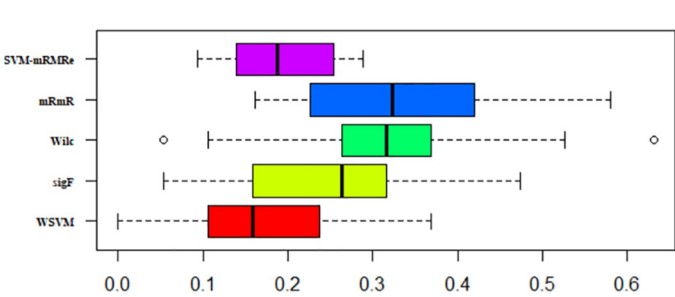

**Fig 10. Box-plots of the error rates produced by random forest, using top 10 features selected by different feature selection methods for Leukemia dataset.**

**Fig 11. Box-plots of the error rates produced by random forest, using top 10 features selected by different feature selection methods for Colon dataset.**

given in Figs 10–17. The boxplots also show that the method, $W_{SNR}$, outperformed the others in majority of the cases.

## 4.1 Simulation

This subsection describes two simulation scenarios for the proposed method. The first scenario ($S_1$) is designed to mimic a situation where the proposed method is useful, whereas the

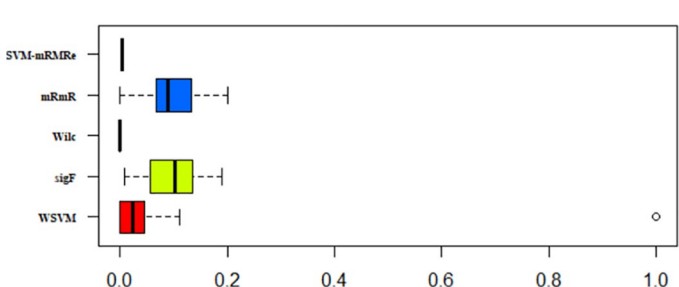

**Fig 12. Box-plots of the error rates produced by random forest, using top 10 features selected by different feature selection methods for Lungcancer dataset.**

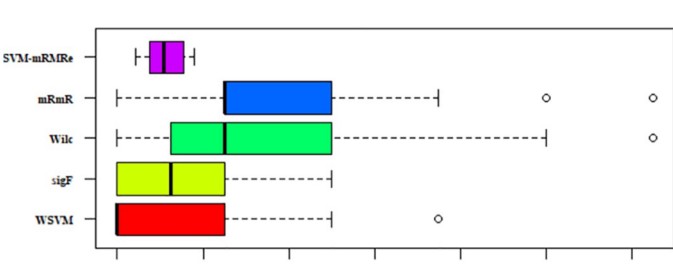

**Fig 13. Box-plots of the error rates produced by random forest, using top 10 features selected by different feature selection methods for Srbct dataset.**

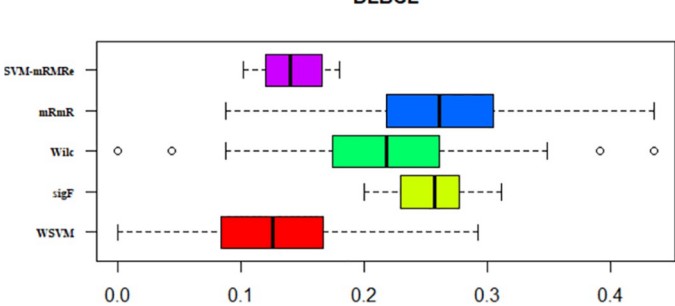

**Fig 14. Box-plots of the error rates produced by random forest, using top 10 features selected by different feature selection methods for DLBCL dataset.**

second scenario ($S_2$) shows a data generation environment that might not favour the proposed method. For this purpose, two different models are designed, one for each scenario. The class probabilities of the Bernoulli response $Y = Bernoulli(p)$ given $n \times d$ dimensional matrix $X$ of $n$ *iid* observations from *Normal*(0, 1) and *Uniform*(0, 1) distributions, are generated in each

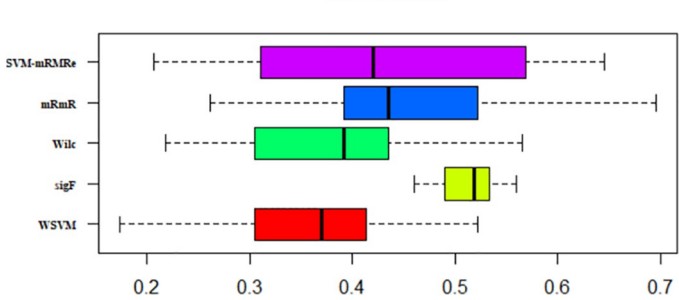

**Fig 15. Box-plots of the error rates produced by random forest, using top 10 features selected by different feature selection methods for Breastcancer dataset.**

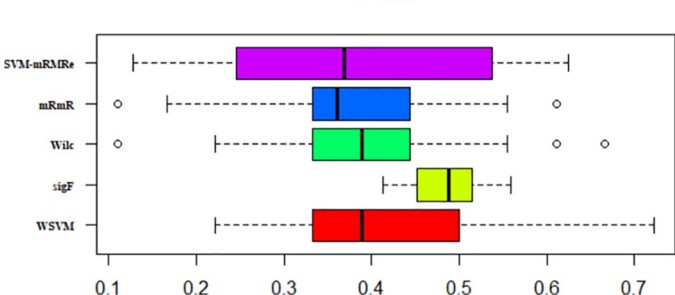

**Fig 16. Box-plots of the error rates produced by random forest, using top 10 features selected by different feature selection methods for TumorC dataset.**

scenario by using the following equation.

$$p(y\|X) = \frac{exp(b \times y + a)}{1 + exp(b \times y + a)}. \tag{9}$$

The values of $a$ and $b$ are both fixed at 1.5. A vector of coefficients, i.e., $\beta$ is generated from the Uniform($-5$, 5) distribution to fit the following linear predictor.

$$Y = X\beta + \epsilon. \tag{10}$$

Top five, i.e., $K = 5$, important variables are identified from the above model based on their coefficients $\beta^s$. In order to contaminate the data, outliers are added to these top five variables from the *Normal*(20, 60) distribution. In addition, 20 noisy variables/observations are also added to the data from *Normal*(5, 10) distribution. By this way a simulated data having $n = 100$ observations and $d = 120$ variables is generated. For all the methods considered, the same experimental set is used as that of the benchmark data sets. The second model is also constructed in a similar fashion. The difference between the two models is that, the former contains outliers and noisy variables/observations in the data, while the latter one does not contain outliers and noisy variables in the data. A total of 500 realizations are made for estimating the performance metrics values. The results of the simulation study for both the scenarios are presented in Table 4.

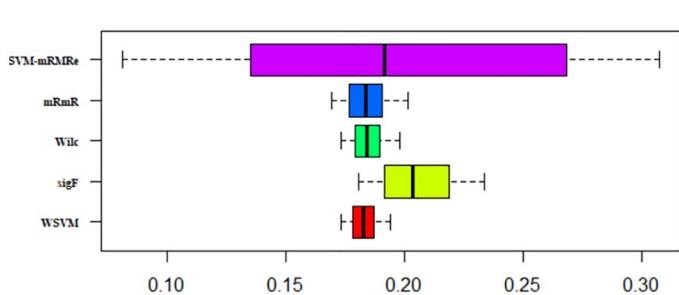

**Fig 17. Box-plots of the error rates produced by random forest, using top 10 features selected by different feature selection methods for Prostate dataset.**

**Table 4. Classification error rates produced by different methods on simulated data.**

| Scenario | Genes | RF | | | | | kNN | | | | |
|---|---|---|---|---|---|---|---|---|---|---|---|
| | | $W_{SNR}$ | sigF | Wilc | mRmR | SVM-mRMRe | $W_{SNR}$ | sigF | Wilc | mRmR | SVM-mRMRe |
| $S_1$ | 5 | **0.227** | 0.388 | 0.296 | 0.336 | 0.320 | **0.299** | 0.401 | 0.449 | 0.349 | 0.352 |
| | 10 | **0.208** | 0.379 | 0.286 | 0.327 | 0.231 | **0.287** | 0.408 | 0.435 | 0.348 | 0.306 |
| | 15 | **0.182** | 0.352 | 0.278 | 0.304 | 0.211 | **0.280** | 0.402 | 0.452 | 0.347 | 0.299 |
| $S_2$ | 5 | 0.474 | 0.421 | **0.339** | 0.417 | 0.440 | 0.505 | 0.429 | **0.299** | 0.422 | 0.302 |
| | 10 | 0.491 | 0.415 | **0.327** | 0.406 | 0.340 | 0.492 | 0.438 | **0.307** | 0.431 | 0.322 |
| | 15 | 0.509 | 0.415 | 0.307 | 0.383 | **0.297** | 0.505 | 0.449 | **0.294** | 0.424 | 0.305 |

From Table 4, it is evident that, when there are noisy variables/observations in the data, the proposed method, $W_{SNR}$, performs better than the other competitors, whereas the method (Wilc) produces minimum error rates, when there are no noisy variables/observations in the data. Similarly, bar-Plots of error rates for different subsets of genes, when the simulated data contains noisy genes/observations in the data and when there are no noisy features/observations are also constructed as given in Figs 18 and 19, respectively. The plots indicate that the

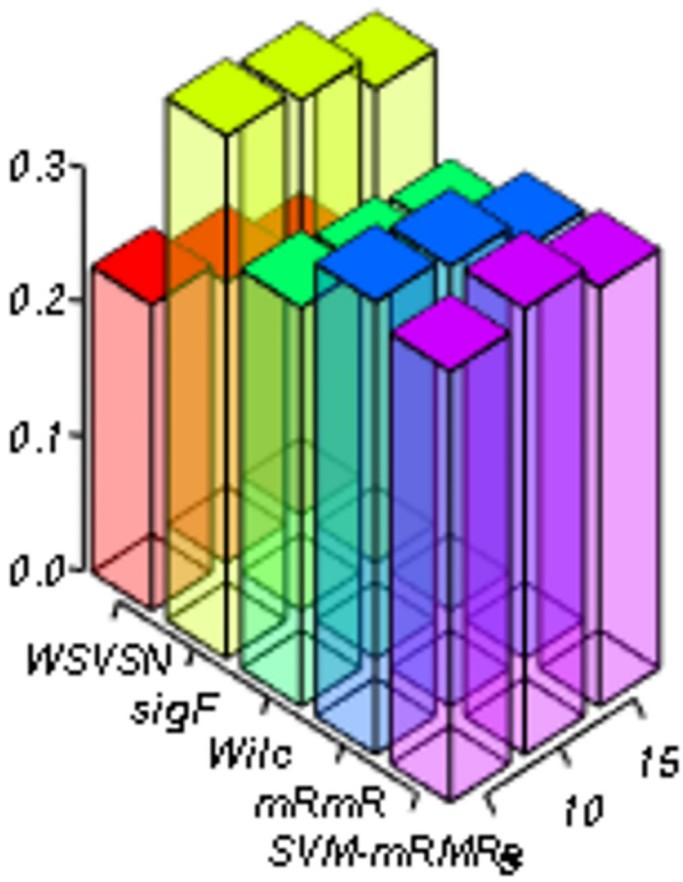

**Fig 18. Bar-plots of errors produced by different feature selection methods on simulated data having outliers, for various subsets of genes.**

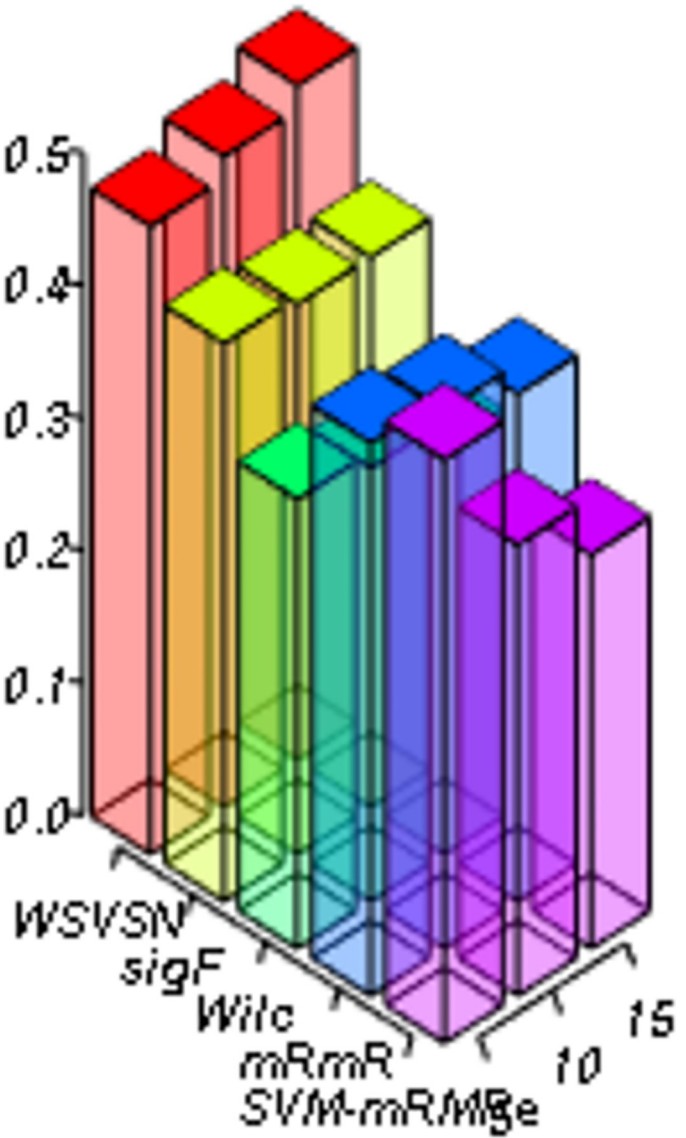

**Fig 19. Bar-plots of errors produced by different feature selection methods on simulated data, having no outliers, for various subsets of genes.**

proposed method, $W_{SNR}$, is producing minimum error rates in the presence of noisy features/observations in the data.

## 5 Conclusion

The current study has proposed a novel feature selection method by exploiting feature weighting via support vectors and signal to noise ratio (SNR). The proposed method initially computes the weights of all genes using support vector machine, followed by the computation of signal to noise ratio for all the genes in the training phase. These weights are then multiplied to compute new weights for each gene in the data. Genes are then arranged in decreasing order of their weights. Top ranked genes are then selected for model construction.

The proposed method is validated on eight benchmark problems and assessment is made against other methods in terms of classification error rates. The results of the proposed method are compared with four well known feature selection methods. Two stat-of-the-art classifiers, i.e., random forest (RF) and $k$-NN are used to evaluate the performance of the selected genes by various feature selection methods. The analyses revealed that the proposed method, $W_{SNR}$, has out performed all the other methods in 6 out of 8 data sets and has produced comparable results on 2 data sets. For quick insight into the results of the proposed method and all the other methods, bar-plots and box-plots of the results have also been constructed. Furthermore, the proposed method is also evaluated on the simulated data where two scenarios are generated. First, a scenario which favors the proposed idea where data consist of noisy features and outlier observations. Second, a scenario where there are no noisy features and outlier observations in the data which does not favor the proposed method. From all the analysis, it is concluded that the proposed method could effectively be used in high dimensional settings where the underlying distribution of observations is not known, as is the case with micro-array data.

For future work in the direction of the proposed study, one can extend it to the situation of unsupervised learning, where the features will first be divided into clusters, and then the proposed method applied in each cluster. The top ranked genes in each cluster can then be selected for the model construction. One can also use the robust measures of location and dispersion in conventional signal to noise ratio to mitigate the effect of outliers in gene expression values. In addition, the performance of the proposed method can be checked by using various kernel functions in SVM.

## Author Contributions

**Conceptualization:** Muhammad Hamraz, Wali Khan Mashwani, Saeed Aldahmani, Zardad Khan.

**Formal analysis:** Muhammad Hamraz, Amjad Ali, Saeed Aldahmani, Zardad Khan.

**Investigation:** Wali Khan Mashwani.

**Methodology:** Muhammad Hamraz, Amjad Ali, Wali Khan Mashwani, Zardad Khan.

**Software:** Amjad Ali, Saeed Aldahmani, Zardad Khan.

**Supervision:** Zardad Khan.

**Validation:** Amjad Ali.

**Visualization:** Amjad Ali.

**Writing – original draft:** Muhammad Hamraz, Amjad Ali, Saeed Aldahmani.

**Writing – review & editing:** Muhammad Hamraz, Wali Khan Mashwani, Saeed Aldahmani.

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
