## [Decision Letter · Decision Letter 0]

15 Dec 2022

PONE-D-22-31389Feature Selection for High Dimensional Microarray Gene Expression Data via Weighted  Signal to Noise RatioPLOS ONE

Dear Dr. Khan,

Thank you for submitting your manuscript to PLOS ONE. After careful consideration, we feel that it has merit but does not fully meet PLOS ONE’s publication criteria as it currently stands. Therefore, we invite you to submit a revised version of the manuscript that addresses the points raised during the review process.

We look forward to receiving your revised manuscript.

Kind regards,

Muhammad Fazal Ijaz

Academic Editor

PLOS ONE

Journal Requirements:

"The authors will pay APC"

"The declare that there is no conflict of interest"

4. Please ensure that you refer to Figure 3 in your text as, if accepted, production will need this reference to link the reader to the figure.

Reviewers' comments:

Reviewer's Responses to Questions

**Comments to the Author**

1. Is the manuscript technically sound, and do the data support the conclusions?

Reviewer #1: Partly

Reviewer #2: Yes

Reviewer #3: Yes

2. Has the statistical analysis been performed appropriately and rigorously? 

Reviewer #1: Yes

Reviewer #2: N/A

Reviewer #3: I Don't Know

3. Have the authors made all data underlying the findings in their manuscript fully available?

Reviewer #1: Yes

Reviewer #2: Yes

Reviewer #3: Yes

4. Is the manuscript presented in an intelligible fashion and written in standard English?

Reviewer #1: Yes

Reviewer #2: Yes

Reviewer #3: Yes

5. Review Comments to the Author

Reviewer #1: The overall impression of the technical contribution of the current study is reasonable. However, the Authors may consider doing necessary amendments to the manuscript for better comprehensibility of the study.

Reviewer #2: Abstract must be rewritten with objectives included and mention accuracy rate of performance.

Write the problem statement.

Mention the contributions made in the paper.

Discuss the Related work.

Justify your approach with the help of performance parameters.

Include Table of Comparison with the state-of-the-art literature.

Proposed method (WSV − Sn) needs to be defined and called it with a precise name.

Reviewer #3: In this paper, authors presented a novel feature selection method by exploiting the weights of features based on support vectors and signal to noise ratio. However, there are some limitations that must be addressed as follows.

1. The abstract is not attractive. Some sentences in abstract should be modified to make it more attractive for readers. In addition, the repetition should be removed (see the word ‘the proposed method’)

2. In Introduction section, it is difficult to understand the novelty of the presented research work. This section should be modified carefully. In addition, the main contribution should be presented in the form of bullets.

3. The authors should discuss the existing work properly. In addition, the following works should be discussed: A smart healthcare monitoring system for heart disease prediction based on ensemble deep learning and feature fusion, An intelligent healthcare monitoring framework using wearable sensors and social networking data, Artificial intelligence in disease diagnosis: a systematic literature review, synthesizing framework and future research agenda, A tri-stage wrapper-filter feature selection framework for disease classification

4. Figures are blurred, their quality should be improved.

5. Captions of the Figures not self-explanatory. The caption of figures should be self-explanatory, and clearly explaining the figure. Extend the description of the mentioned figures to make them self-explanatory.

6. More details should be included in future work.

7. The whole manuscript should be thoroughly revised in order to improve its English.

6. PLOS authors have the option to publish the peer review history of their article (what does this mean?). If published, this will include your full peer review and any attached files.

Reviewer #1: No

Reviewer #2: **Yes: **Dr. Jana Shafi

Reviewer #3: No

---

## [Author Response · Author response to Decision Letter 0]

28 Mar 2023

Dear Editor,

Thank you for allowing us to update our manuscript to be considered in your prestigious journal. The point-to-point response to the reviewers’ comments is as follows:

Reviewer # 1:

Comment: 

The overall impression of the technical contribution of the current study is reasonable. However, the Authors may consider doing necessary amendments to the manuscript for better comprehensibility of the study. 

Response:

Thank you for your feedback. The whole manuscript is revised and all the necessary amendments have been made. We hope the manuscript is now in better position in terms of comprehensibility.

Reviewer # 2:

Comments

Abstract must be rewritten with objectives included and mention accuracy rate of performance. Write the problem statement.

Response:

Thank you for your concern. The abstract is now revised by including objectives and problem statement. Moreover, the performance is evaluated on the basis of error rates, and one can easily determine the accuracy rates of the selected genes, as it is a complement of classification error rate.

Comment # 2: Mention the contributions made in the paper.

Response: 

Thank you for the suggestion. The contribution made in the paper is mentioned by updating the abstract and introduction sections. 

Comment # 3:

Discuss the Related work.

Response:

Related work is now thoroughly discussed.

Response: 

Thank you for your suggestion. The related work is further updated by including some recent and related literature in the field of feature/gene selection in high dimensional gene expression datasets. 

Comment # 5:

Justify your approach with the help of performance parameters.

Response:

Thank you for the suggestion. The method ranks features based on weighted signal to noise ratio and selects the top ranked features. We do not associate any hyper parameter of the method. 

Comment # 6: Include Table of Comparison with the state-of-the-art literature.

Response:

Thank you for your suggestion. The Table of comparison is further updated by including a new feature selection method i.e. (SVM-mRMRe). The results of this new method are updated throughout manuscript. 

Comment # 7: Proposed method (WSV − Sn) needs to be defined and called it with a precise name.

Response:

Thank you for your suggestion. The proposed method is redefined. Moreover, the name (WSV − Sn) is now replaced with precise name.

Reviewer # 3:

Comment # 1: The abstract is not attractive. Some sentences in abstract should be modified to make it more attractive for readers. In addition, the repetition should be removed (see the word ‘the proposed method.

Response: 

Thank you for your concern. The manuscript is updated by doing the required changes.

Comment # 2: In Introduction section, it is difficult to understand the novelty of the presented research work. This section should be modified carefully. In addition, the main contribution should be presented in the form of bullets. 

Response:

Thank you for identifying this flaw. The manuscript is revised by including novelty of the proposed research work. Moreover, the main contribution is now presented in the form of bullets.

Comment # 3: The authors should discuss the existing work properly. In addition, the following works should be discussed: A smart healthcare monitoring system for heart disease prediction based on ensemble deep learning and feature fusion, An intelligent healthcare monitoring framework using wearable sensors and social networking data, Artificial intelligence in disease diagnosis: a systematic literature review, synthesizing framework and future research agenda, A tri-stage wrapper-filter feature selection framework for disease classification.

Response: 

Thank you for your concern. The manuscript is updated by discussing the related work properly. Moreover, the studies suggested by your good self are also incorporated.

Comment # 4: Figures are blurred, their quality should be improved.

Response: Thank you for the suggestions. The quality of figures in the manuscript is now improved.

Comment # 5: Captions of the Figures not self-explanatory. The caption of figures should be self-explanatory, and clearly explaining the figure. Extend the description of the mentioned figures to make them self-explanatory.

Response: Thank you for the suggestion. The manuscript is updated by changing the caption of all figures. The caption are now self-explanatory.

Comment # 6: More details should be included in future work.

Response: Thank you for the suggestion. The required changes have been made and highlighted in the manuscript.

Comment # 7:The whole manuscript should be thoroughly revised in order to improve its English.

Response: Thank you for your feedback . The whole manuscript is revised by improving its language.

---

## [Decision Letter · Decision Letter 1]

5 Apr 2023

Feature Selection for High Dimensional Microarray Gene Expression Data via Weighted  Signal to Noise Ratio

PONE-D-22-31389R1

Dear Dr. Khan,

We’re pleased to inform you that your manuscript has been judged scientifically suitable for publication and will be formally accepted for publication once it meets all outstanding technical requirements.

Kind regards,

Muhammad Fazal Ijaz

Academic Editor

PLOS ONE

Additional Editor Comments (optional):

Reviewers' comments:

Reviewer's Responses to Questions

**Comments to the Author**

1. If the authors have adequately addressed your comments raised in a previous round of review and you feel that this manuscript is now acceptable for publication, you may indicate that here to bypass the “Comments to the Author” section, enter your conflict of interest statement in the “Confidential to Editor” section, and submit your "Accept" recommendation.

Reviewer #1: All comments have been addressed

Reviewer #3: All comments have been addressed

2. Is the manuscript technically sound, and do the data support the conclusions?

Reviewer #1: Yes

Reviewer #3: Yes

3. Has the statistical analysis been performed appropriately and rigorously? 

Reviewer #1: Yes

Reviewer #3: Yes

4. Have the authors made all data underlying the findings in their manuscript fully available?

Reviewer #1: Yes

Reviewer #3: Yes

5. Is the manuscript presented in an intelligible fashion and written in standard English?

Reviewer #1: Yes

Reviewer #3: Yes

6. Review Comments to the Author

Reviewer #1: The authors have addressed all the recommendations of the reviewers in a reasonable manner, manuscript in the current from may be considered for the further phase of editorial process.

Reviewer #3: The authors have addressed my all comments. I have no further comments. Therefore, this paper can be accepted in its present form.

7. PLOS authors have the option to publish the peer review history of their article (what does this mean?). If published, this will include your full peer review and any attached files.

Reviewer #1: No

Reviewer #3: No

---

## [Editor Report · Acceptance letter]

14 Apr 2023

PONE-D-22-31389R1 

Feature Selection for High Dimensional Microarray Gene Expression Data via Weighted Signal to Noise Ratio 

Dear Dr. Khan:

I'm pleased to inform you that your manuscript has been deemed suitable for publication in PLOS ONE. Congratulations! Your manuscript is now with our production department. 

Kind regards, 

on behalf of

Dr. Muhammad Fazal Ijaz 

Academic Editor

PLOS ONE